



# Estimation of the vertical distribution of particle matter (PM2.5) concentration and its transport flux from lidar measurements based on machine learning algorithms

Yingying Ma[1], Yang Zhu[2], Hui Li[1], Shikuan Jin[1], Yiqun Zhang[1], Ruonan Fan[1], Boming Liu[1]*, and Wei Gong[3]*

[1] State Key Laboratory of Information Engineering in Surveying, Mapping and Remote Sensing (LIESMARS), Wuhan University, Wuhan, China
[2] School of Computer Science and Technology, Wuhan University of Science and Technology, Wuhan, China
[3] School of Electronic Information, Wuhan University

*Correspondence to*: Boming Liu (Email: liuboming@whu.edu.cn) and Wei Gong (Email: weigong@whu.edu.cn)

**Abstract.** The vertical distribution of aerosol extinction coefficient (EC) measured by lidar system has been used to retrieve the profile of particle matter with a diameter < 2.5 μm ($PM_{2.5}$). However, the traditional linear model (LM) cannot consider the influence of multiple meteorological variables sufficiently, and then inducing the low inversion accuracy. Generally, the machine learning (ML) algorithms can input multiple features which may provide us with a new way to solve this constraint. In this study, the surface aerosol EC and meteorological data from January 2014 to December 2017 were used to explore the conversion of aerosol EC to $PM_{2.5}$ concentrations. Four ML algorithms were used to train the $PM_{2.5}$ prediction models, including Random Forest (RF), K-Nearest Neighbor (KNN), Support Vector Machine (SVM), and eXtreme Gradient Boosting Decision Tree (XGB). The mean absolute error (root mean square error) of LM, RF, KNN, SVM and XGB models were 11.66 (15.68), 5.35 (7.96), 7.95 (11.54), 6.96 (11.18) and 5.62 (8.27) μg/m$^3$, respectively. This result show that the RF model is the most suitable model for $PM_{2.5}$ inversions from EC and meteorological data. Moreover, the sensitivity analysis of model input parameters was also conducted. All these results further indicated that it is necessary to consider the effect of meteorological variables when using EC to retrieve $PM_{2.5}$ concentrations. Finally, the diurnal and seasonal variations of transport flux (TF) and $PM_{2.5}$ profiles were analyzed based on the lidar data. The large $PM_{2.5}$ concentration occurred at



approximately 13:00–17:00 Location Time (LT) in 0.2–0.8 km. The diurnal variations of the TF shows a clear conveyor belt at approximately 12:00–18:00 LT in 0.5–0.8 km. These results indicated that air pollutants transport over Wuhan mainly occurs at approximately 12:00–18:00 LT in 0.5–0.8 km. The TF near the ground usually have the highest value in winter (0.26 mg/m$^2$ s), followed by the autumn

and summer (0.2 and 0.19 mg/m$^2$ s), respectively, and the lowest value in spring (0.14 mg/m$^2$ s). These findings give us important information of atmospheric profile and provide us sufficient confidence to apply lidar in the study of air quality monitoring.

## 1 Introduction

Aerosol is a suspension of fine solid particles or liquid droplets in air (Hinds, 1999; Chen et al., 2014;

Fan et al., 2019; Huang et al., 2019). In recent decades, with the anthropogenic aerosol emissions increased in China, the concentration of fine particle matter with a diameter of less than 2.5 um (PM$_{2.5}$) in the atmosphere has increased significantly (Ding et al., 2016; Shi et al., 2020; Jin et al., 2021). Moreover, the high concentrations of PM$_{2.5}$ cause haze frequently and reduce atmospheric visibility, directly affecting the ecological environment and human health (Huang et al, 2014; He et al., 2020;

Yin et al., 2021; Raaschou-Nielsen et al., 2013). Besides that, air pollution incidents caused by regional transmission still occur occasionally (Huang et al., 2020; Le et al., 2020). Although the government has taken corresponding environmental protection measures to ensure the gradual desceasing of PM$_{2.5}$, irrational PM$_{2.5}$ concentration control strategies would lead an invalid O$_3$ control and these would hinder O$_3$-PM$_{2.5}$ co-improvements (Liu et al., 2013). Therefore, it is necessary to carry out long-term

continuous monitoring of atmospheric environment, especially the spatial variation characteristics of PM$_{2.5}$ concentrations.

Until now, surface in-situ PM$_{2.5}$ measurements is the most commonly method used by ground stations, due to it can give us the more accurate observations. But the large spatial and temporal variability of PM$_{2.5}$ makes difficult to estimate the abundance at any given locations based upon limited ground





stations (Kumar et al., 2011). Consequently, PM$_{2.5}$ monitoring has been developed from ground-based sampling to satellite or other ground-based remote sensing instruments (Bovouk et al., 2010) gradually, which principle is to obtain the surface PM$_{2.5}$ concentrations from aerosol optical depth (AOD) and meteorological variables. Moreover, it should be stressed in particular that the fine description of

5 atmospheric boundary layer by lidar observations improve the estimation accuracy of surface PM$_{2.5}$ by these instruments (Chu et al., 2013), this is also the preliminary shown of the advantages of lidar profile observation in PM2.5 estimations.

In recent years, transport flux (TF) represents horizontal transmission flux of pollutant is put forward, which is determined by the horizontal wind speed and PM$_{2.5}$ mass concentrations (Tang et al., 2015;

Liu et al., 2019). Obviously, Surface PM$_{2.5}$ observations is not sufficient to reveal the transport of pollutants and the formation process of regional pollution in the whole boundary layer, hence researchers have focused on the vertical distribution of PM$_{2.5}$ mass concentrations (Sun et al., 2013; Zhang et al., 2020; Panahifar et al., 2020). There are three main ways to measure the profiles of PM$_{2.5}$ concentrations. The first is a meteorological tower or unmanned aerial vehicle equipped with PM

detectors, which can directly measure the vertical distribution of PM$_{2.5}$ within the range of 0-0.5 km from the surface (Wu et al., 2009; Yang et al., 2005; Peng et al., 2015). Some high-performance unmanned aerial vehicle can even measure the PM$_{2.5}$ concentrations in the range of 0-1.5 km (Liu, C. et al., 2020). These direct measurement methods have high accuracy, but the detection height is limited to less than 1.5 km. In addition, UAV cannot achieve long-term and uninterrupted observation. The

second way is to use the WRF-Chem model to simulate the vertical profile of PM$_{2.5}$ (Saide et al., 2011; Goldberg et al., 2019; Liu, C. et al., 2021). This way can observe a continuous variation of PM$_{2.5}$ profiles near the surface, while the accuracy of the simulation results need to be improved through field observations. The last method is using lidar or ceilometer to measure the aerosol extinction coefficient (EC) profile, and then retrieve the PM$_{2.5}$ profile based on the EC profile (Lv et al., 2017;

Lyu et al., 2018). Owing to their continuous and large-scale (by changing inclination and



rotating scanning) observation characteristics, lidar and ceilometer are more widely used to monitor the vertical distribution of pollutants in atmosphere (Liu et al., 2018; Liu et al., 2019; Xiang et al., 2021), yet the premise is to construct a suitable conversion model of extinction coefficient to $PM_{2.5}$ mass concentration.

A series studies have been conducted to estimate the $PM_{2.5}$ concentration profile from aerosol EC profile measured by lidar system (Tao et al., 2016; Lyu et al., 2018; Liu et al., 2019; Panahifar et al., 2020). Tao et al. (2016) obtained the vertical distribution of $PM_{2.5}$ mass concentration based on the EC observed by charge-coupled device side-scatter lidar and surface $PM_{2.5}$ concentrations. Lyu et al. (2018) used the EC profile measured by a mobile lidar to retrieved the $PM_{2.5}$ concentration profile in different
seasons at Tianjin. Liu et al., (2019) studied the vertical distribution and TF of $PM_{2.5}$ based on lidar and Doppler wind radar observations. Panahifar et al., (2020) used lidar to give the mass concentrations of dust and non-dust particle in vertical direction when three differences atmospheric environment occur, analysed the influence of local sources pollution from Tehran and long range transported dust from the Arabian Peninsula. These studies retrieved the $PM_{2.5}$ concentration profile by establishing the
linear relationship between aerosol EC and $PM_{2.5}$ concentrations. However, the $PM_{2.5}$ concentrations are not only related to aerosol EC but also related to meteorological factors, such as temperature, relative humidity and wind speed (Bovouk et al., 2010; Chu et al., 2013; Li et al., 2016; Lv et al., 2017). Under the condition that the physical model has been built, the advanced machine learning (ML) techniques offer a possible solution to some nonlinear issues in remote sensing and geoscience fields
(Li et al., 2017). Therefore, the ML algorithms which contain multi-characteristic inputs, have been attempted to be used to estimate the $PM_{2.5}$ concentrations (Chen et al., 2018).

Giving the above mentioned problems and referencing the work of the formers, surface in-situ $PM_{2.5}$, surface aerosol EC and meteorological data from January 2014 to December 2017 were used to explore the conversion model between aerosol EC to $PM_{2.5}$ concentrations. The traditional linear model and
four ML models were used to fit the relationship among surface EC, meteorological parameters and





ground PM$_{2.5}$ concentrations. The performance of linear model and four ML models were then analyzed and compared. After selecting the suitable ML algorithms, in other words, the most effective conversion model can be constructed, finally apply it to the lidar data to obtain the diurnal and seasonal variations of TF and PM$_{2.5}$ profiles during different periods. The rest part of this paper are organized as follows. In sect. 2, the study area and detecting instruments were introduced. The methods for retrieving PM$_{2.5}$ profile were presented in sect. 3. In sect. 4, experiments were conducted, and the experimental results were analyzed. The end of the article, the main findings were summarized.

## 2 Materials and data

### 2.1 Observation station

The observational station is at the State Key Laboratory of Information Engineering in Surveying, Mapping and Remote Sensing (LIESMARS), located at Luoyu road, Wuhan (39.98◦ N, 116.38◦ W), as shown in Fig. 1. The altitude is approximately 23 m above sea level (Liu et al., 2018; Jin et al., 2019). This observational station has been gradually built since 2006, and currently includes a series of equipment such as lidar, nephelometer, aethalometer, particulate matter detector and automatic weather station etc (Zhang et al., 2018; Liu et al., 2018b). In this study, the surface sampling and observation data were used to build conversion models and the performance of model was then contrasted and analysed. The lidar data was used to analyse the vertical distribution of PM$_{2.5}$ concentrations and TF.

### 2.2 Instrumentations and data

#### 2.2.1 Ground-based data

Surface aerosol EC were measured by the combination of nephelometer (Model 3563, TSI, USA) and aethalometer (Model AE31, Magee Scientific, USA). The nephelometer can measure the aerosol



scattering coefficients (SC) simultaneously at 450, 550, and 700 nm, and the error of its data production is less than 7% (Gong et al. 2015). The aerosol SC of lidar at 532 nm can be calculated from wavelengths at 450, 550, and 700 nm (Yan et al., 2017; Liu et al., 2018b). Moreover, the aerosol absorption coefficients (AC) were deduced from black carbon concentrations which were measured

by aethalometer (Xu et al., 2012). The aethalometer can measure the black carbon concentration at the seven wavelengths of 370, 470, 520, 590, 660, 880, and 950 nm. Previous studies indicated that aerosol AC at 532 nm and black carbon concentrations at 880 nm have a strong correlation, and the correlation coefficient ($R^2$) is greater than 0.92 (Yan et al. 2008). Ultimately, the sum of surface aerosol SC and aerosol AC construct the surface aerosol EC. The observation data used for training model were

collected from January 2014 to December 2017.

During this observation period, the particulate matter monitor (Grimm EDM 180, Germany) is used to measure the surface $PM_{2.5}$ concentrations. Moreover, the surface meteorological parameters, such as temperature (T), relative humidity (RH), wind speed (WS) and wind direction (WD) were obtained from an automatic meteorological station (U3－NRC, Onset HOBO, USA). These surface observation

data were processed as hourly averages for matching. After the matching procedure, a total of 5,342 sets of hourly average data were collected.

### 2.2.2 Profile data

A Mie lidar system with an operating wavelength of 532 nm was used to measure the aerosol EC profile. In the measurement, the temporal and spatial resolutions are 1 min and 3.75 m, respectively.

The overlap of this system is 200 m. More detailed descriptions are presented in the previous studies (Liu et al., 2017). This lidar system can directly measure the scattering intensity of aerosols, and aerosol EC can be reversed by the Fernlad method (Fernald, et al., 1984). The Lidar ratio in Wuhan area is supposed as 50 sr (Gong et al., 2010; Liu et al., 2021). The lidar data set includes the observation



from January 2017 to December 2019. After removing the cloud and rain days, a total of 2304 hourly average profiles were obtained.

To calculate the TF of $PM_{2.5}$, the hourly wind profiles were obtained from the fifth generation European Centre for Medium-Range Weather Forecasts atmospheric reanalysis system (ERA-5) (Belmonte et al., 2019). The WS and WD can be calculated from the zonal (u) and meridional (v) component of wind. The wind component data were download from https://cds.climate.copernicus.eu (last access: 13-01-2021) (Liu et al., 2021). In addition, the T and RH profile can also be obtained from ERA-5 data. The wind, T and RH profile data over Wuhan were also download from January 2017 to December 2019 to match the lidar data. Note that the vertical resolution of ERA-5 wind profile is coarser, which only has 12 layers in the height range of 0–3 km. Therefore, for each sample point of ERA-5 data, the lidar data at corresponding height was matched one by one.

## 3 Methodology

In this section, the statistical methods which used to assess the performance of models were first introduced. The establishment of traditional linear model and four ML models was then introduced and discussed. Finally, the calculation method of TF was presented.

### 3.1 Statistical methods

In this study, the mean absolute error (MAE), root mean square error (RMSE), and correlation coefficient (R) were used to assess the performance of each model. Moreover, the MAE was also regarded as an important indicator in the model tuning parameters process. RMSE and MAE are two indexes used in the regression process to represent the difference between predicted and actual values. The lower the variance is, the closer the predicted value is to the actual value. R indicates the





correlation between predicted and actual values. The calculation formulas of MAE, RMSE and R are as follows:

$$\text{MAE} = \frac{\sum_{i=1}^{n}|y_i - x_i|}{n} \tag{1}$$

$$\text{RMSE} = \sqrt{\frac{\sum_{i=1}^{n}(y_i - x_i)^2}{n}} \tag{2}$$

$$R = \frac{\sum_{i=1}^{n}(x_i - \bar{x})(y_i - \bar{y})}{\sqrt{\sum_{i=1}^{n}(x_i - \bar{x})^2}\sqrt{\sum_{i=1}^{n}(y_i - \bar{y})^2}} \tag{3}$$

where $x_i$ and $y_i$ represent the i-th sample point of predicted and actual values, respectively. $\bar{x}$ and $\bar{y}$ represent the mean value of the predicted and actual values, respectively.

*3.2 Traditional linear model*

Traditional linear model (LM) have been used to retrieve the PM$_{2.5}$ mass concentration profile (Lv et al., 2017; Lyu et al., 2018). The physical principle is that the EC is linear with PM$_{2.5}$ when the hygroscopic growth is not considered (Tao et al., 2016). Aerosol EC is composed of SC and AC. Fig. 2 shows the relationship between PM$_{2.5}$ and AC, SC and EC with the variation of RH. The black line represents the fitting result, and the colorbar represents the RH value. For this set of samples, the AC varies from 0 to 0.15, and SC varies from 0 to 1.5. It indicated that the SC of aerosol is dominant. The correlation coefficient (R) between PM$_{2.5}$ and AC, SC and EC were 0.68, 0.8 and 0.82, respectively. The correlation result passed the significance test. These results indicated that the linear model based on SC or EC have the similar performance. This also confirms that the linear model established by SC and PM$_{2.5}$ can also obtain a good inversion results (Liu et al., 2019).

Here, the surface EC and PM$_{2.5}$ concentrations were used to build an LM model. Following Liu et al. method (2019), the linear fitting was restricted through the origin to avoid unreasonable negative values. The red line represents the fitting result after forced passing through the origin (Fig. 2c), and the relationship of LM model is:



$$EC = 0.0067*PM_{2.5} \tag{4}$$

### 3.3 ML methods and optimization

In this study, four classical ML algorithms were used to train a $PM_{2.5}$ prediction model, including Random Forest (RF) (Breiman, 2001), K-NearestNeighbor (KNN) (Altman, 1992; Coomans and Massart, 1982), Support Vector Machine (SVM) (Cao, 2003; Drucker et al., 1997), and eXtreme Gradient Boosting Decision Tree (XGB) (Chen et al., 2015). The input features of these models include EC, RH, T, WD and WS. The total number of experimental samples is 5,342 groups, as mentioned in the Section 2.2.1. Considering the amount of calculation, we randomly pick 90% (4,807) as a training dataset, and the remaining 10% (535) as the testing dataset. Fig. 3 shows that the probability distribution functions (PDF) for training, testing, and whole datasets of observed $PM_{2.5}$ and EC. It is apparent that the PDF of the training dataset (red line) and whole dataset (orange line) are consistent. The testing dataset (blue line) and whole dataset (orange line) have certain deviations in frequency, but the PDF is similar. Previous studies point out that the training dataset with more samples probably do not significantly enhance model performance under a similar distribution (Kühnlein et al., 2014a; Min et al., 2019), therefore, we choose the number of training samples as 4807.

### 3.3.1 Random forest model

RF model is a classifier that uses multiple trees to train and predict samples, which was first proposed by Breiman et al., (2001). There is no correlation between each decision tree in the forest, and the final output of the model is jointly determined by each decision tree in the forest. RF model can handle multiple input features and provide the best outcomes by considering different features. Due to its high degree of generalization and fast training speed, the RF model is widely used in atmospheric remote sensing to solve the nonlinear fitting problem (Wei et al., 2019).





Here, the RF model was used to predict the PM2.5 concentrations, surface EC, RH, T, WD and WS were regarded as inputs. For RF model, three important parameters need to be adjusted to achieve the optimal effect of the model, which include maximum feature (max feature), number of tree (estimator num) and maximum depth of the tree (max depth num), respectively (Table 1). Fig. 4a and 4b show

the tuning parameters process for estimator num and max depth num of RF model under four different max feature. The max feature was set to 0.2, 0.4, 0.6 and 0.8, respectively. The results indicated that the MAE was decreased with max feature increased, while the MAE is almost unaffected when max feature is greater than 0.4. The max feature can be set to 0.4. The values of estimator num and max depth num were then defined at the minimum MAE. After parameter tuning, estimator num and the

max depth num were finally defined to 1000 and 73, respectively.

*3.3.2 K nearest neighbor*

KNN is a ML algorithm that can be used for both classification and regression (Altman, 1992; Coomans and Massart, 1982). Its principle is to find the K training samples closest to it in the training dataset based on the distance metric for a given test sample, and then make predictions based on the

15 information of these K "neighbors". In the atmospheric remote sensing regression task, the average value of the true values of K samples is usually used as the prediction result. Of course, the result of the weighted average based on the distance can also be used as the predicted value (Altman, 1992). The advantage of KNN is that the model can achieve good results in less training time, so it is applied to real-time analysis of some dataset. Due to KNN does not require a model with parameters for

training, only one parameter (number of neighbors) needs to be considered in the optimization of the KNN model. The tuning parameter process for n_neighbors of KNN model was shown in Fig. 4c. According to the curve of MAE changing with n_neighbors, the n_neighbors can be set to 6.





### 3.3.3 Support vector machine

SVM is a two-class classification model, which was first proposed by Cortes and Vapnik in 1995 (Cortes and Vapnik, 1995). Its basic idea is to find a linear classifier with separation hyperplane with maximal interval in the feature space. According to the limited sample information, the best

compromise is sought between the complexity of the model (the learning accuracy of a specific training sample) and the learning ability (the ability to identify any sample without error) in order to obtain the best generalization ability (Drucker et al., 1997). It shows many unique advantages in solving small sample, nonlinear and high-dimensional pattern recognition, and can be extended to other machine learning problems such as function fitting (Cao, 2003).

For SVM model, the penalty parameter (C) and gamma coefficient (g) need to be adjusted to achieve the optimal effect of the model. The tuning parameter process for C of KNN model under four different g was shown in Fig. 4d. The g was set to 0.0001, 0.0003, 0.0005 and 0.0007, respectively. Similarly, it need to take an appropriate C and g value to minimize the MAE. After parameter tuning, the C and g were finally defined to 150 and 0.0005, respectively.

### 3.3.4 Extreme gradient boosting

XGB algorithm is an improved version of Gradient Boosting Decision Tree (GBDT) algorithm. The GBDT algorithm is an additive model that minimizes the objective function value by gradually adding decision trees (Friedman, 2002). However, the objective function does not have a regularization term, it is just the sum of the loss function values, which may easily cause overfitting. The XGB algorithm

adds a regularization term to the cost function on the basis of the GBDT algorithm, and performs a second-order Taylor approximation to the objective function. Then, the exact or approximate method is used to greedily search for the segmentation point with the highest score, and then perform the next segmentation and expand the leaf nodes (Chen et al., 2015). In this way, it is ensured that the tree





structure will not be too complicated to cause overfitting in the process of minimizing the loss function. In addition, this can speed up the calculation.

To achieve the optimal effect of the XGB model, it need to adjust five parameters, including subsample, number of tree (estimator num), maximum depth of the tree (max depth), learning rate and gamma

(Table 1). The tuning parameter process for these parameters was shown in Fig. 5. The subsample was set to 0.1, 0.2, 0.5 and 1, respectively. The results show that subsample=1 is the most suitable. Then according to the change of the green line in each sub-panel, it need to select an appropriate value to minimize the MAE. The estimator num, max depth, learning rate and gamma were finally defined to 400, 6, 0.24 and 0.01, respectively.

*3.4 Calculation method of transport flux*

TF is an important parameter to measure the horizontal transmission of pollutants (Liu et al., 2019; Shi et al., 2020). In this study, the TF is determined by the WS and the PM2.5 concentrations in the area under analysis. The calculation method for a certain height is shown in Eq. (5):

$$TF_i = WS_i * C_i \tag{5}$$

where the $WS_i$ and $C_i$ is the horizontal wind speed and PM2.5 concentrations at a certain height, respectively. According to the profiles of PM2.5 and WS, the TF profile can be obtained.

**4 Results and discussion**

*4.1 Intercomparison of estimated results*

In this section, the estimated PM$_{2.5}$ of LM, RF, KMM, SVM and XGB models were compared and

20 analysed to evaluate the performance of these conversion models. Fig. 6 shows the variation trends of EC, observed PM$_{2.5}$ and the estimated PM$_{2.5}$ by five models. The results indicated that the variation in observed PM$_{2.5}$ was similar to that in the estimated PM$_{2.5}$ of five models. However, it notes that the





observed PM$_{2.5}$ and estimated PM$_{2.5}$ by LM model have a large deviation in sample 1-20. The observed PM$_{2.5}$ were larger than 100 ug/m$^3$, while the corresponding estimated PM$_{2.5}$ of LM was less than 50 ug/m$^3$ (Fig. 6a). This is due to the estimated PM$_{2.5}$ of LM model were directly calculated from EC, resulting to the inaccurate inversion results in some cases. These deviations are improved by machine

learning models, especially in RF and XGB models (Fig.6b and 6c). This is because the ML models consider the influence of meteorological factors such as RH, T, WD and WS. It can be understood that the ML models improve the prediction accuracy through meteorological factor correction. Previous studies have also pointed out that temperature and humidity correction can effectively improve the inversion accuracy of surface PM2.5 (Zhang et al., 2015; Li et al., 2016).

Fig. 7 shows the correlation between the observed PM$_{2.5}$ concentrations and the estimated PM$_{2.5}$ concentration predicted by the five models. The asterisk indicates that correlation coefficient (R) passed the statistical significance difference test ($P < 0.05$). The R of LM, RF, KNN, SVM and XGB models were 0.82, 0.94, 0.87, 0.88 and 0.93, respectively. The MAE (RMSE) of these five models were 11.66 (15.68), 5.35 (7.96), 7.95 (11.54), 6.96 (11.18) and 5.62 (8.27) μg/m$^3$, respectively. These

results show that these four ML algorithms had a better fitting effect, and the error was only half of the LM error. It indicated that the performance of ML algorithms is obvious better than that of LM algorithm. Among the four ML algorithms, RF and XGB models have similar performance, and both are better than KNN and SVM models. The RF model have the highest R and the smallest MAE. It shows that the RF model is the most suitable model for PM$_{2.5}$ inversion based on the EC.

*4.2 Sensibility analysis*

From the results in previous section, the ML algorithms that takes meteorological variables into account has better performance than the LM algorithm. The input variable importance analysis was performed to investigate the influence of meteorological factors, as shown in Fig. 8. For these four model, the importance ranking of the input variables is same, which is EC, WD, WS, T and RH. But





there is a large difference in the importance value of each input variable. The importance value of EC in RF, KNN, SVM and XGB are 0.51, 0.87, 0.71, and 0.66, which is much larger than other input features. It indicated that the concentration of PM$_{2.5}$ was main affected by EC. This also proves that the surface EC and PM$_{2.5}$ have a very good linear relationship when the RH is less than 70% (Tao et al., 2016; Lv et al. 2017). Another special point is that the importance value of RH is approximately 0.10 in RF and XGB models, while the effect of RH can be ignored in KNN and SVM models. Combined with the results in Fig.7, it finds that the models which considered the effect of aerosol moisture absorption growth have a better performance. In addition, the effect of WS and T are also ignored in KNN model. This leads the performance of KNN model weaker than the that of other three models. These results indicated that it is necessary to consider the effect of meteorological variables when using EC to retrieve PM2.5 concentrations.

Fig. 9 shows the difference between estimated and observed PM$_{2.5}$ that changed with EC. The gray, red, green, blue and orange points represent the difference between LM-observed, RF-observed, KNN-observed, SVM-observed, and XGB-observed, respectively. The black line indicated the frequency of difference. For LM model, most of the estimated PM$_{2.5}$ is overestimated when the EC is larger than 0.6. This may be due to that the LM model does not take into account the influence of humidity. The heavy pollution weather is usually accompanied by higher humidity, and the hygroscopic growth effect of aerosols cannot be ignored (Zhang et al., 2015; Liu et al., 2018). By contrast, the difference between estimated and observed PM$_{2.5}$ is smaller in the ML models. In these four models, the frequency with a difference of less than 5 ug/m$^3$ can reach 0.68, 0.47, 0.59, and 0.65, respectively. The frequency of difference in four ML models is similar. Moreover, the deviation of the ML models is relatively stable and does not change with the increase of EC. It also notes that although five meteorological variables are input in the ML model, not all models take into account the influence of each parameter, which leads to differences in the performance of the model. Overall, the performance of RF and XGB models are better than SVM and KNN models.



*4.3 Vertical evolution of PM$_{2.5}$ and TF*

In this section, the diurnal and seasonal variations of TF and PM$_{2.5}$ profiles were analyzed during different periods in Wuhan. Due to the best performance of RF model, the PM$_{2.5}$ profiles were retrieved based on the RF model.

Fig. 10 shows the diurnal variation of the EC, WS, PM$_{2.5}$ and TF profiles. The daily maximum value of the EC appeared at approximately 08:00–13:00 local time (LT) in 0.4–0.6 km. The EC below 1 km has obvious diurnal characteristics, which is larger during the daytime (08:00–20:00 LT) and smaller at nighttime (Fig. 10a). By contrast, the WS below 1 km is larger during the nighttime and smaller at daytime. The daily minimum value of WS occurred at approximately 13:00–17:00 LT in 0.2–1 km

(Fig. 10b). For the diurnal variation of PM$_{2.5}$, the high PM$_{2.5}$ concentrations at nighttime is mainly concentrated below 0.5 km. After sunrise (08:00 LT), the PM$_{2.5}$ concentrations increased, and the pollution layer is higher in the vertical direction, distributed between 0.2-0.8 km. The diurnal variations of TF profiles were similar with that of PM$_{2.5}$ profiles (Fig. 10d). At near ground, the peak TF was 0.26 mg/m$^2$ s and then remain at approximately 0.15 mg/m$^2$ s. There was an obvious conveyor belt at

approximately 12:00–18:00 LT in 0.5–0.8 km. These results indicated that the transport of pollutants over Wuhan mainly occurred between 12:00 and 18:00 LT, which was similar to the results of previous studies (Ge et al., 2018; Liu et al., 2019).

Fig. 11 shows the seasonal variation of the PM$_{2.5}$ and TF profiles. The concentration of PM$_{2.5}$ at 0.2 km has the highest value in the winter (93.7 ug/m$^3$), followed by the autumn and summer (80.3 and

75.8 ug/m$^3$, respectively), and lowest in the spring (53.5 ug/m$^3$). This finding is similar to the surface observation results (Wang et al., 2016). The PM$_{2.5}$ concentration decreases gradually with the height increases. The PM$_{2.5}$ concentration decreases rapidly in the height range of 0.2 to 1 km, but the rate of reduction has obviously seasonal differences. The decline rate of the PM$_{2.5}$ in the winter and autumn is higher than that in the spring and summer. An interesting phenomenon is that the PM$_{2.5}$ mass



concentrations during summer is large in the height range of 0.6 to 1.5 km. This may be caused by the transmission of dust in summer (Liu et al., 2018; 2020). The vertical profiles of the TF is similar to that of $PM_{2.5}$ concentrations (Fig. 11e-h). The seasonal mean TF at 0.2 km is the highest in winter (0.26 mg/m$^2$ s), followed by the autumn and summer (0.2 and 0.19 mg/m$^2$ s, respectively), and lowest

5  in spring (0.14 mg/m$^2$ s). With the height increasing, the TF profiles has obvious seasonal difference. The variations in the spring and autumn are similar, the TF gradually decreases with the height increases. In the summer (Fig. 11f), the TF is approximately 0.19 mg/m$^2$ s in the height range of 0.2 to 0.5 km, and then declines above 0.5 km. The decrease rate above 0.5 km is slower than other seasons. In the winter (Fig. 11h), the TF is stable (approximately 0.26 mg/m$^2$ s) in the height range of 0.2 to 0.5

10  km, and declines rapidly above 0.5 km. These results indicate that the transport of pollutants mainly occurs in 0.2–1 km. In general, in the autumn and winter, the TF and $PM_{2.5}$ concentrations are concentrated near the ground, indicating that local emissions are the main source of $PM_{2.5}$ (Zhang et al., 2021). In the summer, the TF is relatively high in 0.5–1.5 km, indicating that the concentration of $PM_{2.5}$ over Wuhan is affected by high-altitude dust transport (Tao et al., 2013; Liu et al., 2020). In the

15  spring, the TF and $PM_{2.5}$ concentrations are at a low level, indicating that the air quality in Wuhan area is better in spring.

## 5 Summary and conclusions

This study presents a comprehensive analysis to explore the conversion of aerosol extinction coefficient to $PM_{2.5}$ concentrations based on the surface observation data from January 2014 to

20  December 2017. The correlation and difference between observed and estimated $PM_{2.5}$ have been analysed to evaluate the performance of LM, RF, KNN, SVM and XGB models. Furthermore, diurnal and seasonal variations of TF and $PM_{2.5}$ profiles have been investigated.





After using traditional LM and other four ML algorithms to predict the $PM_{2.5}$ mass concentrations profile. The R of LM, RF, KNN, SVM and XGB models were 0.82, 0.94, 0.87, 0.88 and 0.93, respectively. The MAE (RMSE) of these five models were 11.66 (15.68), 5.35 (7.96), 7.95 (11.54), 6.96 (11.18) and 5.62 (8.27) μg/m$^3$, respectively. These results show that the RF model is the most

suitable model for $PM_{2.5}$ estimations. Moreover, the importance value of EC in RF, KNN, SVM and XGB models are 0.51, 0.87, 0.71, and 0.66, respectively. It proved that EC plays an important role in $PM_{2.5}$ estimations. The frequency with a difference of less than 5 ug/m$^3$ were 0.30, 0.68, 0.47, 0.59, and 0.65 in the LM, RF, KNN, SVM and XGB models, respectively. Combined with the importance value of input variables, the results indicated that the conversion models which considers the effect of

meteorological variables has the smallest deviation. Finally, the diurnal and seasonal variations of TF and $PM_{2.5}$ profiles were analysed. For diurnal variations, the high $PM_{2.5}$ concentrations at nighttime is mainly concentrated below 0.5 km.  At daytime, the pollution layers usually suspend in the higher altitude, and distribute between 0.2-0.8 km. The high TF appeared at approximately 12:00–18:00 LT in 0.5–0.8 km. These results indicated that the transport of pollutants over Wuhan mainly occurred

between 12:00 and 18:00 LT. For seasonal variations, the TF and $PM_{2.5}$ mass concentrations are concentrated near the ground in autumn and winter, indicating that local emissions are the main source of $PM_{2.5}$ during these periods. In the summer, TF has the relatively high value in 0.5–1.5 km, which indicates the concentration of $PM_{2.5}$ over Wuhan is affected by high-altitude dust transport.

Our work comprehensively compares the performance of LM, RF, KNN, SVM and XGB models.

From the perspective of correlation and deviation between observed and estimated $PM_{2.5}$, we conclude that the performance of RF and XGB models are better than others, followed by SVM and KNN models, last is LM model. This information can provide us a reference to apply lidar data in air quality research.



**Data availability**

The experimental data used in this paper can be provided for non-commercial research purposes upon request (Dr. Boming Liu: liuboming@whu.edu.cn). The ERA5 wind data can be download from https://cds.climate.copernicus.eu/cdsapp#!/dataset/reanalysis-era5-pressure-levels?tab=form (last

accessed: 24 May 2021). Instructions for use and data access methods can be found on the official website.

**Author contributions**

The study was completed with close cooperation between all authors. Y. Ma and B. Liu conceived of the idea for the manuscript; Y. Ma and B. Liu conducted the data analyses and co-wrote the manuscript;

Y. Zhu, H. Li, S. Jin, W. Gong, Y. Zhang, and R. Fan discussed the experimental results, and all coauthors helped reviewing the manuscript.

**Competing interests.**

The authors declare that they have no conflict of interest.

**Acknowledgements.**

We are very grateful to the lidar team of Wuhan university for the operation and maintenance of the lidar and ground-based instruments. This work was financially supported by the LIESMARS Special Research Funding, National Natural Science Foundation of China under grants 42001291 and Project funded by the China Postdoctoral Science Foundation 2020M682485.

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



## Tables:

**Table 1.** Summary of tuning parameters and their dynamic ranges of four different machine learning algorithms

| Algorithm | Parameter | Dynamic range |
|---|---|---|
| RF | 1. maximum feature (max feature) | [0.2, 0.4, 0.6, 0.8] |
|  | 2. number of tree (estimator num) | [0–1400 within an interval of 10] |
|  | 3. maximum depth of the tree (max depth num) | [10–590 within an interval of 1] |
| KNN | 1. number of neighbors (n neighbors) | [0–25 within an interval of 1] |
| SVM | 1. penalty parameter (C) | [0–1000 within an interval of 50] |
|  | 2. gamma coefficient (g) | [0.0001, 0.0003, 0.0005, 0.0007] |
| XGB | 1. subsample | [0.1, 0.2, 0.5, 1] |
|  | 2. number of tree (estimator num) | [0–480 within an interval of 20] |
|  | 3. maximum depth of the tree (max depth), | [1–20 within an interval of 1] |
|  | 4. learning rate | [0.01–0.5 within an interval of 0.01] |
|  | 5. gamma | [0.01–0.99 within an interval of 0.02] |



## Figures:

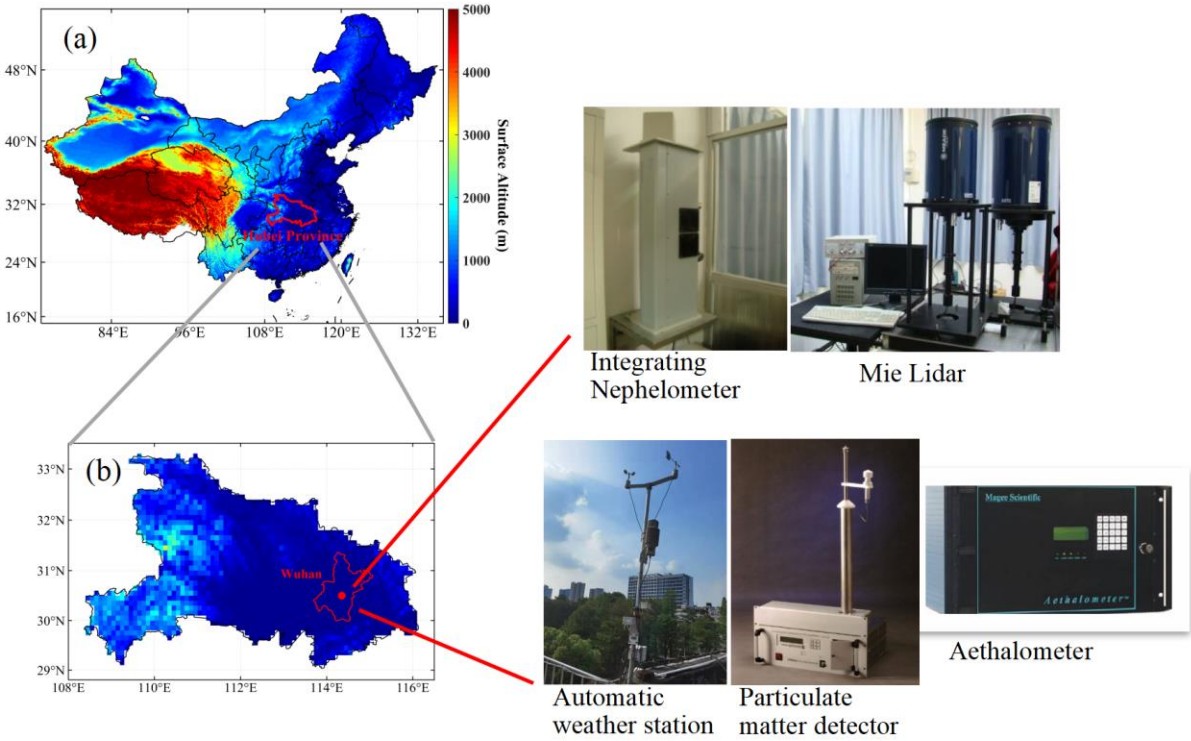

**Figure 1.** Geographic distribution of observation site and the observation instruments used in this

5    study. The photo of particulate matter detector is provided by GRIMM Aerosol Techink (@ GRIMM)





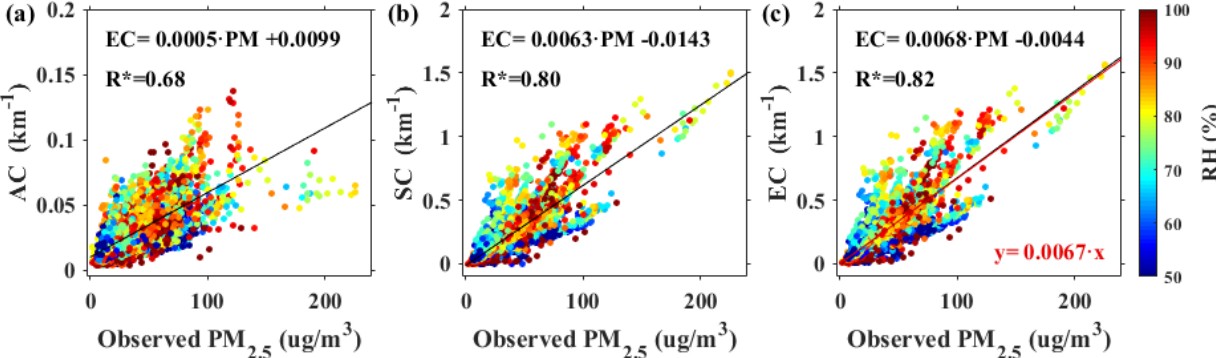

**Figure 2.** The linear regression relationship between observed PM$_{2.5}$ and (a) AC, (b) SC, (c) EC with the change of RH. The black line is the regression line, and the red line is the regression line through the origin. The color bar represents the RH.





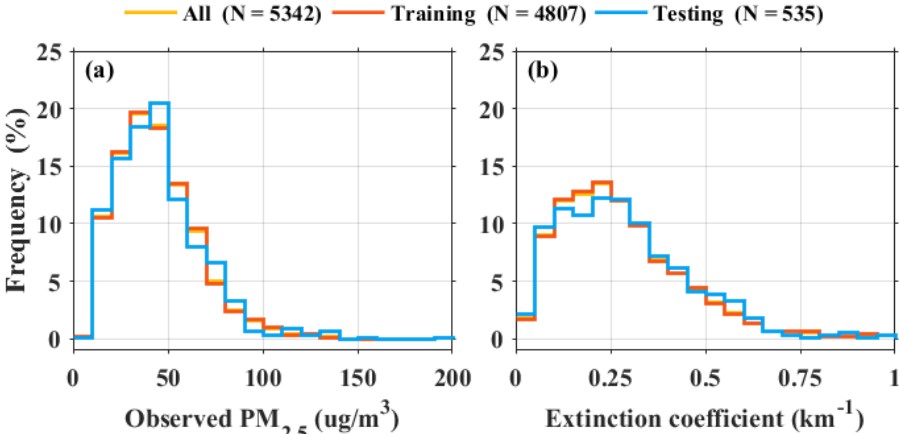

**Figure 3.** Probability distribution functions of all sample datasets (orange line), training dataset (red line), and testing (blue line) for observed (a) PM$_{2.5}$ and (b) EC. N represents the total number of samples of every dataset.





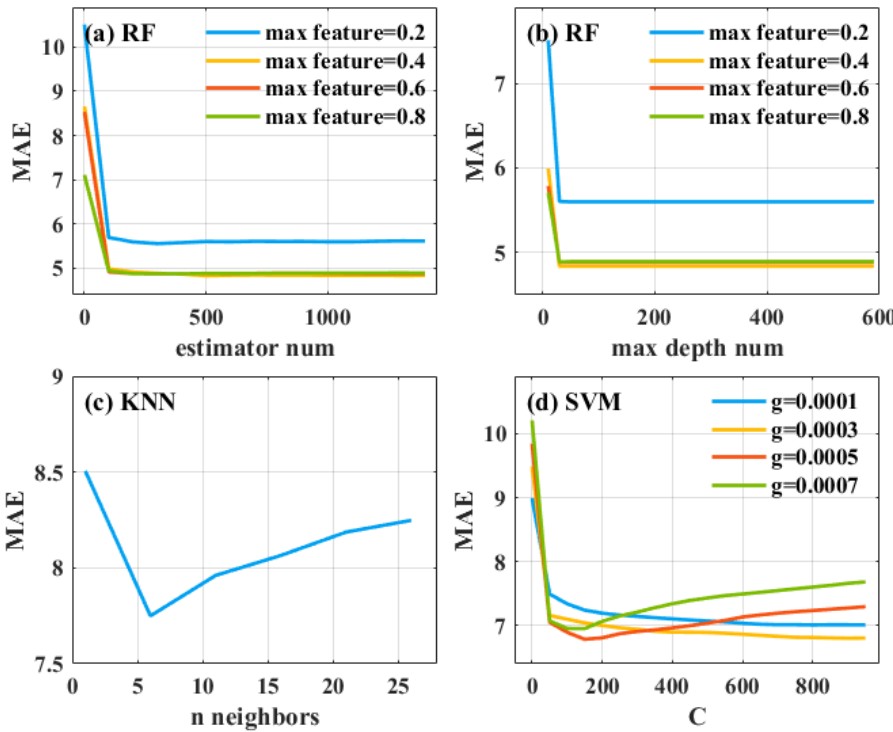

**Figure 4.** Mean absolute errors (MAE) between observed PM$_{2.5}$ and estimated PM$_{2.5}$ based on the (a, b) RF, (c) KNN, and (d) SVM models under different tuning process.



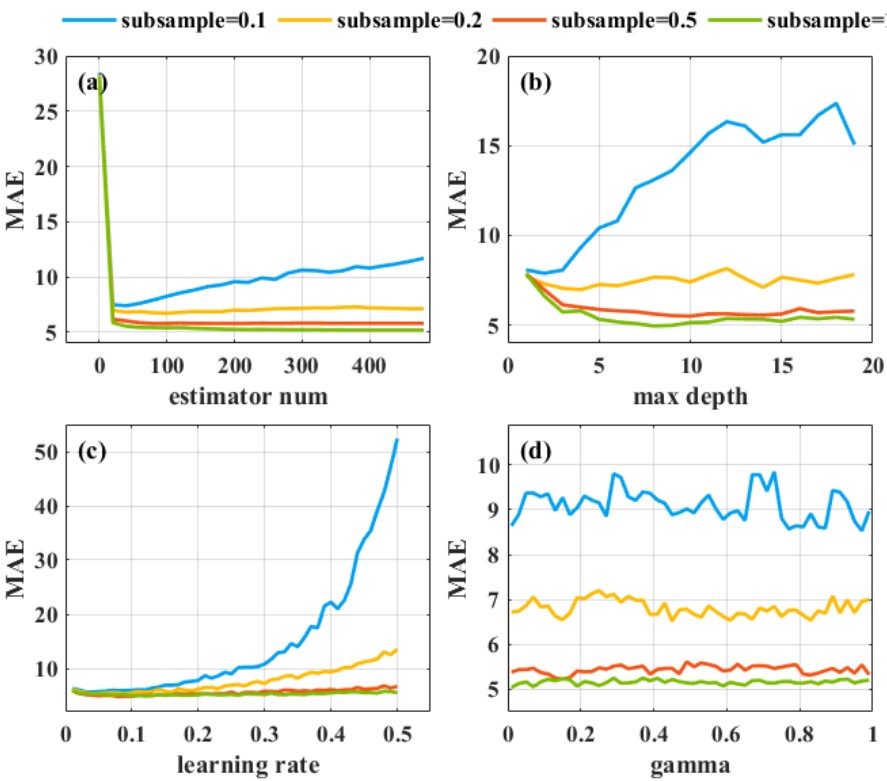

**Figure 5.** Mean absolute errors (MAE) between observed PM$_{2.5}$ and estimated PM$_{2.5}$ based on the XGB algorithms under the tuning process of (a) estimator num, (b) max depth, (c) learning rate and (d) gamma. Blue, orange, red and green lines indicate the subsample under different value.

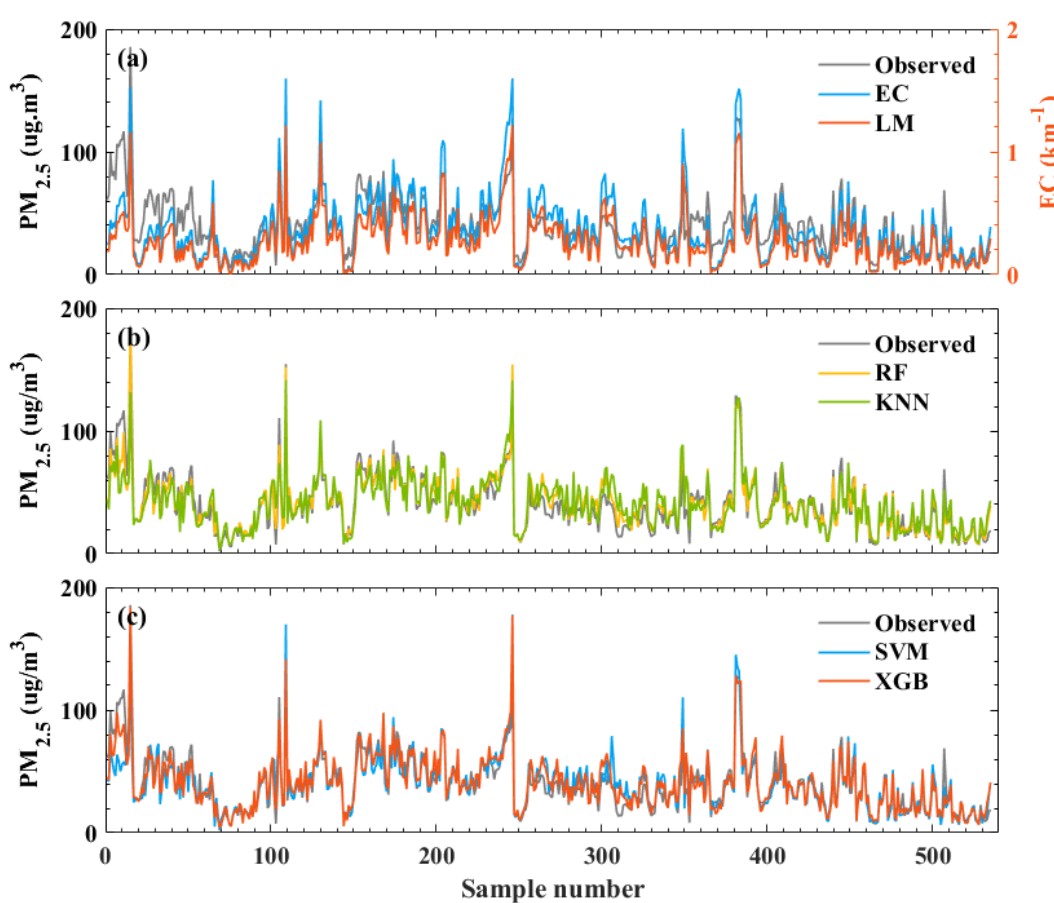

**Figure 6.** Variations of the estimated PM$_{2.5}$ predicted by (a) LM, (b) RF and KNN, (c) SVM and XGB. The gray line represents the observed PM$_{2.5}$.



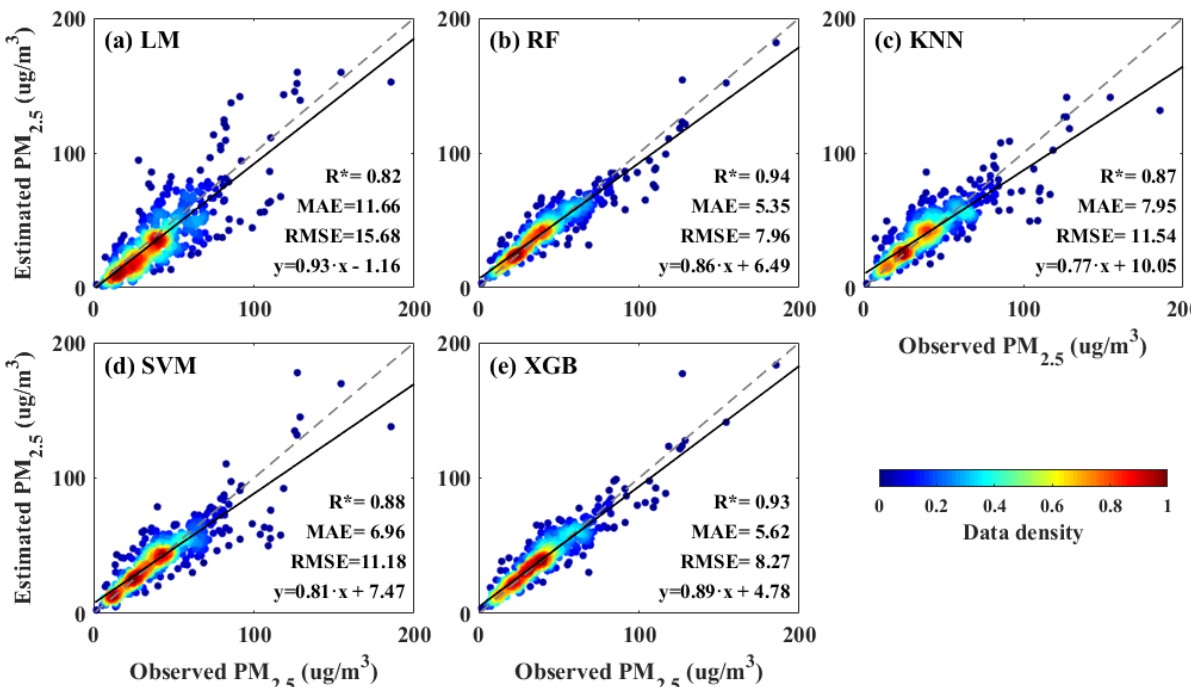

**Figure 7.** Correlation coefficients between observed PM$_{2.5}$ and estimated PM$_{2.5}$ based on the (a) LM, (b) RF, (c) KNN, (d) SVM and (e) XGB models. The gray and black line is the reference and regression line, respectively. The asterisk indicates that correlation coefficient (R) passed the statistical significance difference test (P < 0.05).



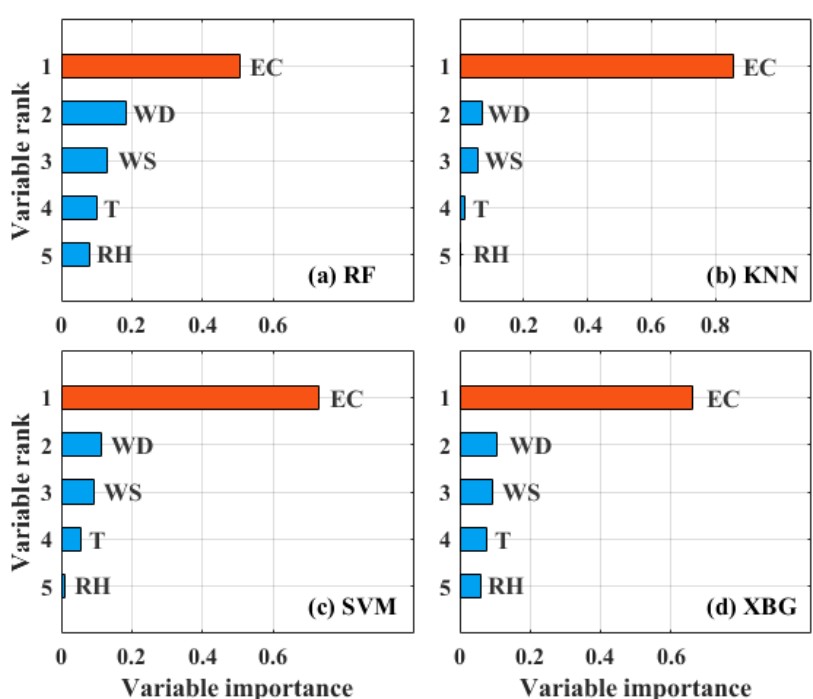

**Figure 8.** Ranking histograms of the input environment variable for (a) RF, (b) KNN, (c) SVM and (d) XGB models.



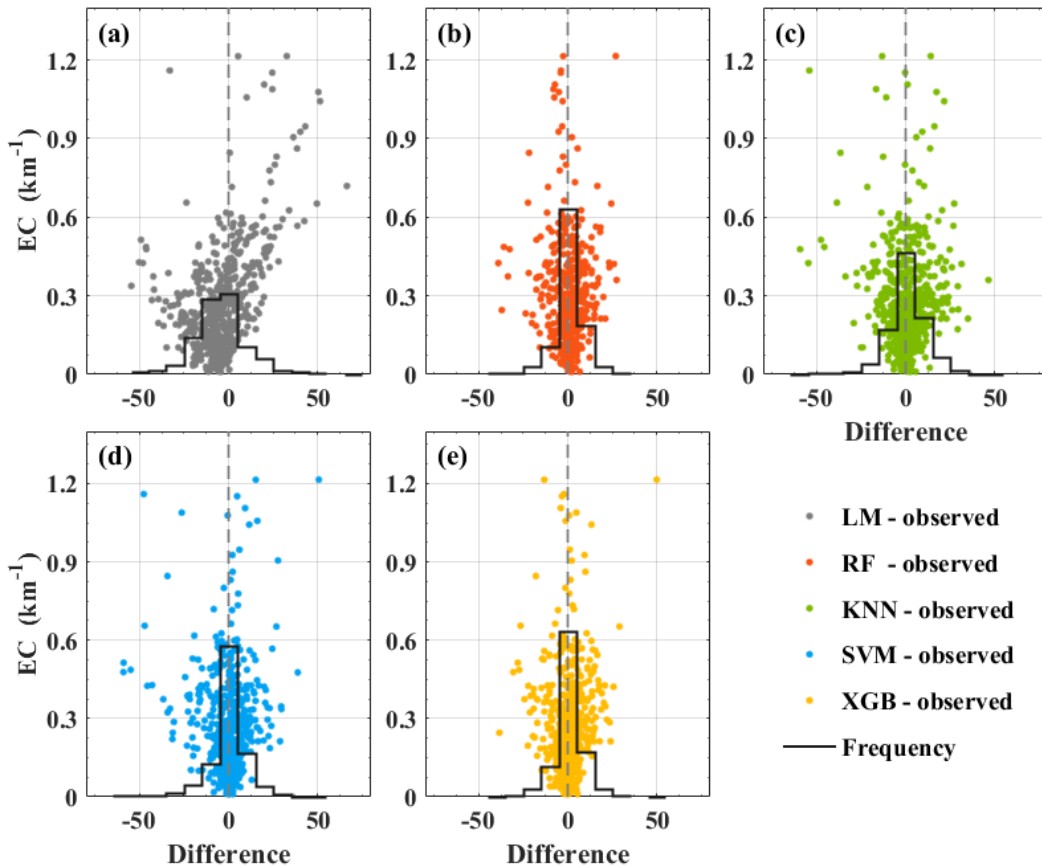

**Figure 9.** Difference of observed PM$_{2.5}$ and estimated PM$_{2.5}$ with the change of EC for (a) LM, (b) RF, (c) KNN, (d) SVM and (e) XGB models. The gray, red, green, blue and orange points represent the difference between LM-observed, RF-observed, KNN-observed, SVM-observed, and XGB-observed, respectively. The black line represents the frequency.





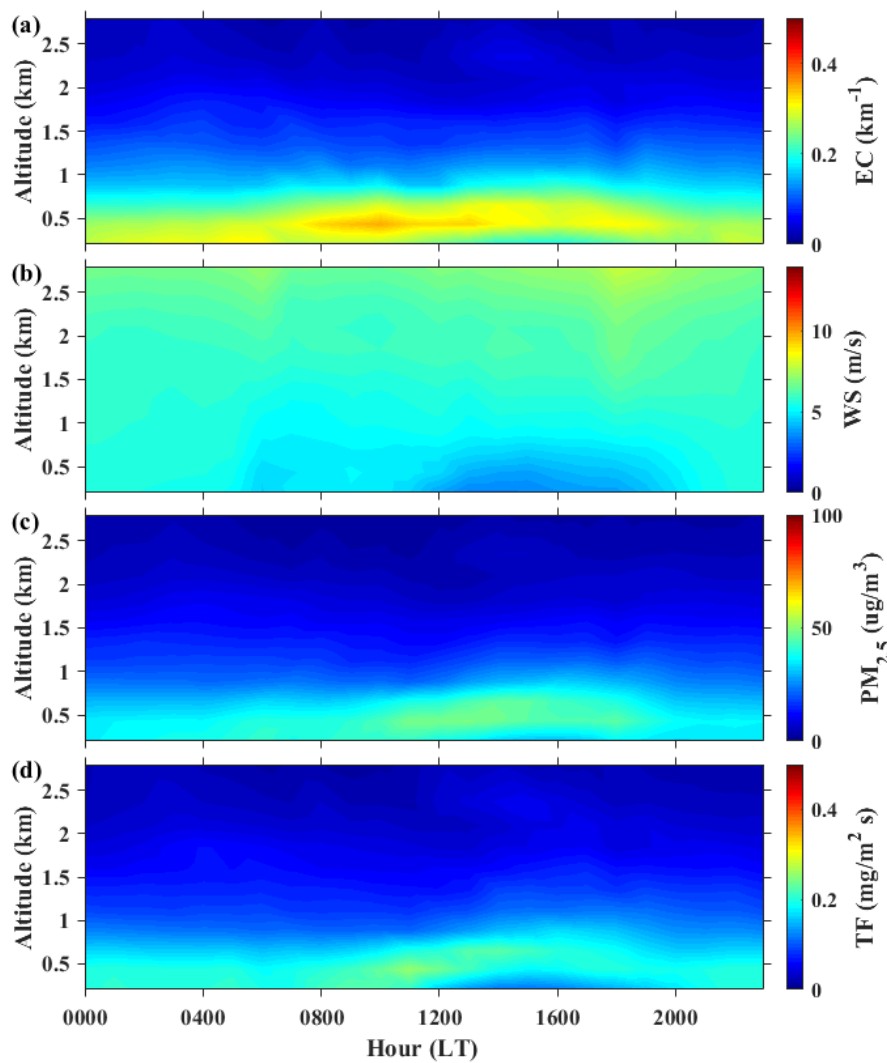

**Figure 10.** Hourly variations of vertical distribution of (a) EC, (b) WS, (c) PM$_{2.5}$ and (d) TF in Wuhan from January 2017 to December 2019.



**Figure 11.** Seasonal and annual profiles of (a, b, c, d) PM$_{2.5}$ and (e, f, g, h) TF from January 2017 to December 2019. Corresponding color-shaded areas represent standard deviation.