# Peer review of "Estimation of the vertical distribution of particle matter (PM2.5) concentration and its transport flux from lidar measurements based on machine learning algorithms"

_Atmospheric Chemistry and Physics, 2021_

## Author Comment (AC1)

**Response to Reviewer #1's Comments**

Vertical distribution of PM2.5 and its flux are of great importance to evaluate its impacts on environment, climate as well as human health. The article proposes a new method for estimating vertical distribution of PM2.5 concentrations from active remote sensing observation based on ML algorithms. The topic is of sufficient interest to the communities of study of atmospheric aerosol and environments. In general, I find this manuscript to be of interest for publication and appropriate for Atmospheric Chemistry and Physics. There are several suggestions for improvement listed below that should be considered by the authors and the editors before publication.

***Response: We thank the anonymous reviewer for his/her comprehensive evaluation and thoughtful comments, which greatly improve the quality of our manuscript. We have made efforts to adequately address the reviewers' concern one by one. For clarity purpose, here we have listed the reviewer' comments in plain font, followed by our response in bold italics.***

P3, line 17, the abbreviation of "unmanned aerial vehicle" should be added.

***Response: Amended as suggested.***

P6, Line 7, is the R2 the correlation coefficient or determination coefficient? Please confirm it.

***Response: Good question! The R2 should be the determination coefficient. We have modified it in text.***

P8, Line 16, please clarify the level of significance test.

***Response: Amended as suggested. The level of significance test is P<0.05. We have added it in text.***

P9, Line 8-9, "we randomly pick 90% (4,807) as a training dataset, and the remaining 10% (535) as the testing dataset." I am confused the method of picking the training and testing dataset, please focused two questions: 1ï¼‰After multiply and randomly picking samples, do the so-called remaining 10% samples participate in the model training? Or are the 10% not involved in the training of the model at all times? If the

first one, this means that the predictive performance these models are unreliable. Please give the detailed explanation.

*Response: Good question! In here, the remaining 10% data was regarded as the independent testing dataset. The testing dataset are not involved in the training of the model;it is only used to evaluate model performance. We have added it in text.*

About the validation of model training in section 3.3 and the evaluation of predictive power in section 4.1, the authors should consider more methods, e.g. sample-based 10-fold cross-validation.

*Response: Good suggestion! As your said, 10-fold cross-validation is a good way for the validation of model training. Considering the amount of calculation, we follow min et al.'s (2020) method. We randomly pick 90% as a training dataset, and the remaining 10% as the independent testing dataset. We think this method can be used well for model training. Therefore, we did not use more methods for validation.*
*Reference: Min, M., Li, J., Wang, F., Liu, Z., & Menzel, W. P. (2020). Retrieval of cloud top properties from advanced geostationary satellite imager measurements based on machine learning algorithms. Remote Sensing of Environment, 239, 111616.*

P12, Formula (5), please give the unit of transport flux in the corresponding context so as to understand it conveniently, because the unit of transport flux at a certain height is different from that of column-integrated transport flux.

*Response: Amended as suggested. The unit of transport flux is $ug/m^2$ s. We have added it in text.*

P13, Line 24, model àmodels

*Response: Amended as suggested.*

Section 5 should be rewritten. This section is just repeating some statements that have been made in the previous sections. In a good conclusion, the authors should interpret all the findings and even discussion with a higher level of abstraction.

***Response: Good suggestion! According to your suggestion, we rewrite the Section 5. "After using traditional LM and other four ML algorithms to predict the PM$_{2.5}$ mass concentrations profile. The results show that the performance of ML algorithms is better than traditional LM algorithm. This is due to the ML models consider the effect of meteorological variables, and can conduct the temperature and humidity correction to improve the inversion accuracy. Moreover, for the four ML algorithms, the RF model is the most suitable model for PM$_{2.5}$ estimations, followed by XGB model, last are SVM and KNN models. The difference in model performance is due to the difference in the decision tree structure of the model. Each ML algorithm has its own decision-making method to consider the weight of input parameters. Combined with the importance value of input variables and the deviation of results, the results indicated that the higher weight of the meteorological parameters in the model, the smaller deviation of the results."***

---

## Author Comment (AC2)

**Response to Reviewer #2's Comments**

In this paper, authors want to Estimation of the vertical distribution of particle matter (PM2.5) concentration based on machine learning algorithms, that's a good idea. This is a problem worth studying. Before publication, there are some problems to pay attention to.

*Response: We greatly appreciated the reviewer's positive comments on our manuscript, which greatly improve the quality of our manuscript. We have made efforts to adequately address the reviewers' concern one by one. For clarity purpose, here we have listed the reviewer' comments in plain font, followed by our response in bold italics.*

1 ¼Œthe author get the aerosol EC profile from a mie lidar, using a Lidar ratio as a Constant hypothesis, which is 50. if you can give an Error caused by constant assumption, it will be better.

*Response: Good suggestion! According to the previous study (Liu et al., 2017), the standard deviation of the assumed lidar ratio is about 20%, the uncertainty for EC derived by lidar is about 10%-20%. We have added it in the text.*

*Reference: Liu, B., Ma, Y., Gong, W., & Zhang, M. (2017). Observations of aerosol color ratio and depolarization ratio over Wuhan. Atmospheric Pollution Research, 8(6), 1113-1122.*

2, in part 5 Summary and conclusions, please Refine the summary part, and give some constructive discussion ¼Œ it will be better.

*Response: Good suggestion! According to your suggestion, we rewrite the Section 5. "After using traditional LM and other four ML algorithms to predict the PM2.5 mass concentrations profile. The results show that the performance of ML algorithms is better than traditional LM algorithm. This is due to the ML models consider the effect of meteorological variables, and can conduct the temperature and humidity correction to improve the inversion accuracy. Moreover, for the four ML algorithms, the RF model is the most suitable model for PM2.5 estimations, followed by XGB model, last are SVM and KNN models. The difference in model performance is due to the difference in the decision tree structure of the model. Each ML algorithm has its own decision-making method to consider the weight of input parameters.*

*Combined with the importance value of input variables and the deviation of results, the results indicated that the higher weight of the meteorological parameters in the model, the smaller deviation of the results.*"